# Schnitzler Syndrome: Insights into Its Pathogenesis, Clinical Manifestations, and Current Management

**DOI:** 10.3390/biom14060646

**Published:** 2024-05-31

**Authors:** Antoine Braud, Dan Lipsker

**Affiliations:** Dermatologic Clinic, University Hospital of Strasbourg, 67091 Strasbourg, France; dan.lipsker@chru-strasbourg.fr

**Keywords:** Schnitzler syndrome, autoinflammation, NLRP3 inflammasome, monoclonal gammopathy, interleukin 1-beta, anakinra

## Abstract

Schnitzler syndrome is a rare disorder characterized by a chronic urticarial rash associated with immunoglobulin M (IgM) monoclonal gammopathy. Schnitzler syndrome shares strong clinicopathologic similarities with monogenic IL-1-mediated autoinflammatory disorders and is now considered an acquired adult-onset autoinflammatory disease. The spectacular effect of interleukin-1 inhibitors demonstrates the key role of this cytokine in the pathogenesis of the disease. However, the physiopathology of Schnitzler syndrome remains elusive, and the main question regarding the relationship between autoinflammatory features and monoclonal gammopathy is still unanswered. The purpose of this narrative review is to describe what is currently known about the pathogenesis of this peculiar disease, as well as to address its diagnosis and management.

## 1. Introduction

Schnitzler syndrome is a rare acquired autoinflammatory disorder that was first described in 1972 [1]. It is characterized by a chronic urticarial rash associated with immunoglobulin M (IgM) monoclonal gammopathy. Other features include recurrent fever, arthralgia, and bone pain associated with abnormal bone remodeling. The average age of onset of Schnitzler syndrome is 50 to 55 years with a slight male predominance. Very few cases have been published with onset before the age of 35. Most reported patients are of Caucasian descent, but cases have been described in more than 25 countries around the world [2].

Severe impairment of quality of life and AA amyloidosis used to be the major complications of persistent inflammation. Treatment with IL-1-blocking therapies has dramatically improved the care of Schnitzler patients by minimizing the risk of these complications. The prognosis is mostly related to the potential development of a B-cell lymphoproliferative disease, most commonly Waldenström’s macroglobulinemia.

The physiopathology of Schnitzler syndrome is still largely unknown, and the relationship between the autoinflammatory features and the IgM paraprotein has not been elucidated.

Here, we aimed to recapitulate what is currently known about the pathogenesis of Schnitzler syndrome, as well as to describe its diagnosis and treatment.

## 2. Pathogenesis

### 2.1. Schnitzler Syndrome as an Autoinflammatory Disease

Autoinflammatory diseases are a group of disorders characterized by abnormal activation of the innate immune system leading to sustained systemic inflammation.

Schnitzler syndrome shares strong clinicopathologic similarities with cryopyrin-associated periodic syndrome (CAPS). CAPS is a monogenic IL-1-mediated autoinflammatory disorder caused by a gain-of-function mutation in the gene NOD-like receptor family, pyrin domain containing 3 (*NLRP3*), which encodes a protein called NLRP3 or cryopyrin.

#### 2.1.1. The NLRP3 Inflammasome

NLRP3 is an intracellular pattern-recognition receptor that recognizes pathogen-associated or danger-associated molecular patterns (PAMPs and DAMPs). Upon stimulation, NLRP3 proteins oligomerize, recruit an adaptor protein called ASC, and assemble into a single macromolecule called an ASC speck. The assembled ASC recruits and activates caspase 1. Activated caspase 1 cleaves biologically inactive pro-IL-1β and pro-IL-18 to produce mature cytokines. The cleavage of gasdermin D leads to its insertion into the membrane, forming pores that can lead to pyroptosis [3].

Activation of the NLRP3 inflammasome is regulated by a two-step process. A Toll-like receptor (TLR)-mediated priming signal activates the nuclear factor-κB (NF-κB) pathway, which upregulates NLRP3 and pro-IL-1β expression. Interestingly, IL1-1β can act as a priming stimulus upon binding to its receptor, IL-1R. NLRP3 is then activated by a second signal upon recognition of a wide variety of PAMPs or DAMPs [3].

CAPS encompasses a continuous spectrum of phenotypes of varying severity, including Familial Cold Autoinflammatory Syndrome (FCAS), Muckle–Wells Syndrome (MWS), and Neonatal-Onset Multisystem Inflammatory Disease (NOMID). All of these diseases are caused by a gain-of-function mutation of *NLRP3*, which is responsible for protein self-polymerization leading to spontaneous formation of the NLRP3 inflammasome and uncontrolled production of IL-1β. The main symptoms of CAPS are an urticarial rash, a recurrent fever, and arthralgia.

#### 2.1.2. No Evidence for NLRP3 Mutation in Schnitzler Syndrome

Schnitzler syndrome shares many clinical and biological features with CAPS. Somatic mutations of *NLRP3* have previously been implicated in the pathogenesis of Schnitzler syndrome, and myeloid lineage mosaicism of *NLRP3* has been reported in two patients with Schnitzler-like syndrome [4]. These patients, who did not have IgM gammopathy, were likely cases of late-onset mosaic CAPS.

No somatic or germline variations of *NLRP3* were identified by deep next-generation sequencing (NGS) in two large cohorts of 21 and 40 patients [5,6]. In addition, the same studies did not identify any other pathogenic variations in either NGS panel of genes known to be involved in systemic autoinflammatory disease.

#### 2.1.3. Schnitzler Syndrome Is an IL-1-Mediated Disease

Despite the absence of mutation in *NLRP3*, several lines of evidence link Schnitzler syndrome to IL-1β and activation of the NLRP3 inflammasome. The dramatic and immediate effect of IL-1 inhibition is probably one of the strongest indications of the major role of this cytokine in the pathogenesis of the disease.

Rowczenio et al. reported that patients with Schnitzler syndrome had elevated levels of IL-18 and IL-6 compared to healthy controls [5]. While IL-1β is usually barely detectable in plasma, elevated IL-18 levels suggest caspase-1 activation, as its precursor requires cleavage. In the same study, the authors showed elevated levels of extracellular ASC specks, which are known to be released during NLRP3-mediated pyroptosis [3]. Krause et al. reported a low but elevated level of IL-1β in Schnitzler patients compared to healthy controls [7]. Hold et al. reported increased phosphorylation of interleukin-1 receptor-associated kinase 4 (IRAK-4) in both B cells and monocytes in a patient with Schnitzler syndrome. IRAK-4 acts as a signal transducer for TLR, IL-1R, and IL-18R. Phosphorylated IRAK-4 was reduced to near-normal levels after treatment with an IL-1 inhibitor (anakinra). Notably, no increase in STAT3 phosphorylation was detected, suggesting no upregulation of IL-6 signaling. Regnault et al. showed that peripheral blood mononuclear cells (PBMCs) from Schnitzler patients spontaneously released higher levels of IL-1β, IL-1α, IL-6, and TNFα compared to healthy controls, and that lipopolysaccharide (LPS) stimulation increased the production of these cytokines [8]. Interestingly, the latter study also identified changes in the adaptive immune response. After stimulation of PBMCs with anti-CD3/CD28 beads, the production of IFNγ, IL-4, IL-17A, and IL-10 was lower in patients compared to healthy controls, suggesting possible T-cell immunosuppression in Th1, Th2, Th17, and Treg functions.

If Schnitzler syndrome is caused by a defective inflammasome, it remains unknown what causes this dysregulation. While dermal neutrophilic infiltration is a major feature in the active lesional skin of Schnitzler patients, it has been suggested that dermal mast cells are the main source of IL-1β [9], similar to what has been described in CAPS [10]. IL-1 secretion is likely to be one of the causes of neutrophil recruitment into the dermis. Dermal neutrophils are often seen with leukocytoclasis (i.e., the fragmentation of neutrophil nuclei into dust). Activated neutrophils have the ability to release neutrophil extracellular traps (NETs) composed of nuclear components and granules. This process, called NETosis, has been identified in the lesional skin and blood of patients with Schnitzer syndrome and patients, as well as in patients with neutrophilic dermatoses [11,12]. The NETosis rate of healthy control neutrophils after stimulation with a phorbol ester (PMA) was increased in the presence of symptomatic patient serum compared to healthy control serum, suggesting that factors in the serum, probably cytokines such as IL-1β or IL-6, may enhance this phenomenon [11]. The exact role of NETs in autoimmune or autoinflammatory diseases is unknown, as they have been shown to both stimulate and downregulate the inflammatory response [13,14]. CCL2 is another chemoattractant identified by Krause et al. in the serum of patients with Schnitzler syndrome [7]. CCL2 was also shown to be produced by both PBMCs and dermal fibroblasts upon IL-1 β stimulation and may contribute to the recruitment of mononuclear immune cells to various organs, including skin and bone tissue.

The underlying cause of IL-1 β upregulation is likely to be multiple, resulting in an altered balance between inflammasome activation and negative feedback mechanisms. While no genetic alteration has been identified in genes known to be involved in autoinflammatory diseases, including *NLRP3*, a somatic variant in the *MYD88* gene (*L265P*) has been described in several patients with Schnitzler syndrome [15]. *MYD88* encodes a downstream adaptor protein involved in both the TLR and IL-1R pathways. This gain-of-function mutation triggers NF-κB signaling by genocopying the conformational effects of activating phosphorylation [16]. This MYD88 mutation could lead to a persistent NF-κB priming signal, upregulating both NLRP3 and Il-1 β. In addition, negative feedback regulation of MyD88 signaling by caspase-1-mediated cleavage has recently been described, and the *L265P* variant has been shown to be resistant to this caspase-1-mediated inhibition [17]. Of interest, the *MYD88 L265P* mutation is found in more than 90% of Waldenström disease patients and may be one of the links between IL-1 inflammation and the IgM paraprotein (see below) [18].

### 2.2. Schnitzler Syndrome as a Monoclonal Gammopathy of Clinical Significance

The presence of a monoclonal gammopathy is an obligate criterion of Schnitzler syndrome. Monoclonal IgM gammopathy is largely predominant, mainly associated with a kappa light chain. The Strasbourg criteria (see below) allow the inclusion of IgG gammopathies that are much less common, while some cases associated with IgA gammopathy have been published [19]. At the very beginning of the disease, the monoclonal component can be present at a very low level, while high IgM levels may suggest Waldenström disease. The main question that remains to be answered is whether the paraprotein is the cause or the consequence of the disease process.

The IgM paraprotein was initially thought to be the cause of the skin lesion. It has been shown that IgM deposits can be detected in the skin with the same isotype as the monoclonal gammopathy, but at different sites and with different antigen targets [20]. More recently, Pathak et al. performed deep sequencing of the immunoglobulin heavy chain from 10 patients and a protein microarray using isolated IgM from 3 patients: both analyses failed to identify a shared B-cell clonality [21].

Delayed detection of IgM paraprotein up to 4 years after symptom onset has been reported [22]. These results do not support the assertion that the IgM paraprotein is the original causative agent in the pathogenesis of Schnitzler syndrome. On the contrary, descriptions of remission of Schnitzler syndrome symptoms after treatment with chemotherapy (cyclophosphamide +/− rituximab), though rare, could argue for a pathogenic role of gammopathy [23,24].

The second hypothesis is that the paraprotein may be the consequence of prolonged inflammation inducing plasma cell clonality. The delayed detection of IgM mentioned above could be explained by this mechanism. IL-6, which is consistently elevated in Schnitzler patients, is known to be a growth and survival factor in plasma cell dyscrasia [25]. However, other IL-1-mediated diseases such as CAPS, which have a similar inflammatory profile, are not associated with monoclonal gammopathy. In addition, while treatments that inhibit IL-1 are dramatically effective on the symptoms of Schnitzler syndrome, they have no effect on the monoclonal component and do not prevent the development of lymphoproliferative disease.

The third hypothesis could be that the paraprotein is neither the cause nor the consequence but shares a similar pathogenesis with inflammasome dysregulation. The MYD88 *L265P* mutation described above could be one of these shared mechanisms. This mutation was identified in a subset of Schnitzler patients and is found in most cases of Waldenström’s macroglobulinemia. This mutation leads to an increase in NF-κB signaling, which is known to play a central role in plasma cell dyscrasia. Of note, in IgM monoclonal gammopathy of unknown significance (MGUS), the *MYD88* mutation predicted progression to Waldenström macroglobulinemia with hazard ratios greater than 20 [18]. However, since this mutation is not consistently identified in Schnitzler syndrome patients, it is only one possible player among many other unknown ones that may also be associated with increased activity of the NF-κB pathway.

## 3. Clinical Features

### 3.1. Rash

In most cases, a recurrent urticarial rash is the first presenting symptom, which may precede others by months or even years. This peculiar rash consists of pink to red, barely raised papules or plaques with no change in the skin surface. The lesions are monomorphic, confluent, and located on the limbs and trunk, often sparing the face, palms, and soles. Individual lesions last less than 24 h and resolve without scarring. A halo of vasoconstriction and dermographism may be seen, but angioedema is rare. In contrast to classic urticaria, pruritus is usually absent. Mild itching or burning may develop over time in a subset of patients. The frequency of eruptions is variable, ranging from daily to several times a year. Antihistamines are ineffective.

Histopathologic examination reveals a neutrophilic urticarial dermatosis [26]. The dermis contains a neutrophilic perivascular and interstitial infiltrate with leukocytoclasis. There is little to no edema and no vasculitis. The perivascular neutrophilic infiltrate associated with leukocytoclasis should not be confused with vasculitis, as there are no fibrinoid changes in the vessel walls in neutrophilic urticarial dermatosis. Neutrophilic epitheliotropism is highly suggestive, especially around the sweat glands. This histopathologic pattern of neutrophilic urticarial dermatosis is not specific to Schnitzler syndrome and may be seen in CAPS, adult-onset Still’s disease, or lupus erythematosus.

### 3.2. Recurrent Fever

Recurrent fever is present in a majority of patients. The body temperature can rise above 40 °C but is usually well tolerated without chills. Fever flares can be accompanied by a rash or musculoskeletal pain. Fever occurs without a periodic pattern, with a variable frequency ranging from daily to a few times a year.

### 3.3. Musculoskeletal Involvement

Musculoskeletal involvement is another feature, affecting more than two-thirds of patients. Joint pain is common, but arthritis is exceptional and should cast some doubt on the diagnosis. Joint involvement is not erosive, and joint destruction is not a feature of Schnitzler syndrome. Bone pain is a characteristic finding, most commonly affecting the lower limbs (tibia, femur, or pelvis), although pain may also occur in the spine, forearm, or clavicle.

Objective identification of skeletal involvement is important in the diagnosis of Schnitzler syndrome. Radiographic abnormalities have been reported prior to the onset of clinical symptoms. None of the imaging abnormalities are specific, as they may be seen in other dysplastic or infiltrative diseases. Sclerotic lesions in Schnitzler syndrome can have a similar aspect to those in Erdheim–Chester disease (a non-Langerhans cell histiocytosis), systemic mastocytosis, POEMS, sclerotic myeloma, or endochondroma [27].

Osteosclerosis is the most frequent radiological finding, but mixed lesions with both lysis and sclerosis may be seen. Bone scintigraphy is more sensitive than conventional radiology in detecting lesions. In a cohort of 18 patients, increased tracer uptake was seen in 83% of patients, with a median number of bone lesions of nine and with 3 patients having a unique lesion [28]. The most common sites were the femur and tibia, followed by the humerus, radius and ulna, fibula, and pelvic bones. Involvement of the distal femur and proximal tibia has been referred to as the “hot knee” sign, which is equivocal for both Schnitzer syndrome and Erdheim–Chester disease [27]. Interestingly, patients treated with anti-IL1 were associated with a marked improvement in bone scan abnormalities, with complete resolution in some patients [28]. Magnetic resonance imaging can also show sclerosis that can be associated with medullar bone involvement ranging from a mild patchy medullary signal to extensive medullar edema [27]. Lesions can also be visualized using FDG-PET, showing an increase in FDG uptake in sites of bone sclerosis.

When performed, pathologic examination is often normal or may show nonspecific sclerosis or nonspecific inflammation [2]. Biologically, patients with Schnitzler syndrome have been shown to have elevated blood markers of bone formation (bone-specific alkaline phosphatase and osteocalcin, both produced by osteoblasts) without markers of bone resorption (normal levels of CTX and sRANKL) [29]. The causes of the increase in osteoblast function remain unknown, as IL-1 is known to inhibit osteoblasts, and both IL-1 and IL-6 are stimulators of osteoclasts. The increase in bone formation may be related to an IL-1-mediated increase in angiogenesis, as circulating VEGF levels were found to be elevated in untreated patients and significantly decreased with IL-1 inhibition [29].

### 3.4. Organomegaly

Enlarged lymph nodes can be found in about 25% of patients, usually in the axilla or groin. They may be multiple, permanent, and suggestive of a lymphoproliferative disorder. Lymph node biopsy shows reactive lymphadenitis. Hepatomegaly and splenomegaly are found in a small number of patients [26].

### 3.5. Other Clinical Signs

Asthenia, weight loss, and myalgia are common clinical signs. Neuropathy has also been reported in a minority of patients, usually in the form of a symmetrical sensory polyneuropathy. A case of Schnitzler syndrome associated with aortitis, with both conditions responding immediately to IL1 blockade, has been reported [30]. Pancreatitis has been reported in one case of Schnitzler syndrome [31], but the pancreatitis preceded other symptoms by 15 years and the patient had a clear family history of pancreatitis, making it unlikely to be linked with Schnitzler syndrome.

### 3.6. Biological Findings

As mentioned above (see *Pathogenesis*), the presence of a monoclonal gammopathy is a mandatory criterion for Schnitzler syndrome. Schnitzler syndrome is mainly associated with monoclonal IgM gammopathy with a kappa light chain (85% of cases) [2]. Monoclonal IgG gammopathy is less common, and only a few cases of monoclonal IgA gammopathy have been described, either isolated or associated with an IgM component [19,32].

Inflammatory markers such as C-reactive protein (CRP) or erythrocyte sedimentation rate (ESR) are almost always elevated. A complete blood count often reveals neutrophilic leukocytosis, while inflammatory anemia with thrombocytosis may occur secondary to chronic and persistent inflammation. As mentioned above, alkaline phosphatase levels may be elevated due to abnormal bone remodeling. Complement levels are normal or elevated, in contrast to hypocomplementemic urticarial vasculitis or cryoglobulinemic vasculitis.

## 4. Diagnosis

### 4.1. Diagnostic Criteria

The first diagnostic criteria were established by Lipsker et al. in 2001 (Table 1) [33].

These criteria were updated in 2012 at an expert meeting in Strasbourg [34], with the main difference being the inclusion of IgG monoclonal gammopathy in addition to IgM (Table 2).

Both diagnostic criteria were later validated in 2017 in a cohort of 42 already-diagnosed patients [35]. It is important to remember that the reliability of these criteria has not been evaluated in recent-onset disease. Patients may not fully meet all criteria at presentation. In the case of a neutrophilic urticarial dermatosis associated with an IgM monoclonal gammopathy, Schnitzler’s syndrome must be suspected, even if the minor criteria are not sufficient.

### 4.2. Differential Diagnosis

Both chronic spontaneous urticaria (i.e., not neutrophilic urticarial dermatosis) and monoclonal gammopathy are not uncommon, especially in the elderly. Schnitzler’s syndrome should not be overdiagnosed in the absence of minor criteria. Other diseases that may be associated with urticarial rash, fever, or systemic symptoms include the following:-Urticarial vasculitis is characterized by fixed and purpuric lesions, usually lasting more than 24 h. Histopathologic examination reveals vasculitis (i.e., swelling of endothelial cells, extravasation of erythrocytes, and fibrinoid necrosis of small vessel walls). Complement is decreased in the hypocomplementemic variant.-Cryoglobulinemic vasculitis, which is also characterized by a purpuric rash with vasculitis on skin biopsy, hypocomplementemia, and the presence of cryoglobulinemia.-Adult-onset Still’s disease, often associated with initial pharyngitis, abnormal liver function tests, and (very) high ferritin levels with low glycosylated ferritin.-Genetic autoinflammatory syndromes such as CAPS, which usually present at an early age. Patients with low-grade CAPS mosaicism can present with neutrophilic urticarial dermatosis and an increase in markers of inflammation, but they usually lack the monoclonal IgM component (D. Lipsker, personal observation).

## 5. Treatment

Prior to the use of IL-1 inhibitors, Schnitlzer syndrome was a difficult disease to treat. Several treatments have been used, but none have been able to induce a durable remission, except for inappropriately high and prolonged doses of steroids [36]. IL-1 inhibitors have dramatically changed the management of these patients. They are now the first-line treatment for Schnitzler syndrome, with high efficacy, a rapid response, and few side effects.

### 5.1. IL-1 Blockade Therapy

#### 5.1.1. Anakinra

Anakinra is a recombinant human interleukin-1 receptor antagonist (IL-1RA). It has the same amino acid sequence as native IL-1RA with the addition of an N-terminal methionine residue. It acts as a competitive inhibitor by binding to IL-1R, thereby inhibiting the biological activity of IL-1α and IL-1β. Anakinra has been approved by the U.S. Food and Drug Administration (FDA) for the treatment of rheumatoid arthritis, Neonatal-Onset Multisystem Inflammatory Disease (NOMID), the most severe form of CAPS, and more recently for the treatment of deficiency interleukin-1 receptor antagonist (DIRA). In Europe, anakinra is also approved for the treatment of Familial Mediterranean Fever and Still’s disease.

The efficacy of anakinra in Schnitzler syndrome was first reported in 2005 [37], and it has since become the main treatment for Schnitzler syndrome. Anakinra is administered subcutaneously and has a half-life of 3 to 9.5 h. Symptoms resolve within hours of injection, but relapse usually occurs within 24–48 h if treatment is not continued. Therefore, injections given daily or every other day are usually required to maintain remission. Most patients respond to anakinra, and anakinra itself serves as a diagnostic test. Resistance to anakinra should prompt a review of the diagnosis.

There are no major contraindications to anakinra except hypersensitivity to the drug. Anakinra is usually well tolerated with the exception of frequent injection site reactions. The neutrophil count should be assessed before starting the treatment and then monitored as neutropenia has been described with anakinra. Rare cases of serious infections or hepatitis have been reported.

#### 5.1.2. Canakinumab

Canakinumab is a human anti-IL-1β monoclonal antibody. It has been approved by the FDA for the treatment of several periodic fever syndromes (CAPS, Familial Mediterranean Fever, Tumor Necrosis Factor Receptor-Associated Periodic Syndrome [TRAPS], Hyperimmunoglobulin D Syndrome [HDS], and Mevalonate Kinase Deficiency [MKD]), as well as Still’s disease and gout.

Compared to anakinra, canakinumab has a longer half-life of 22.9 to 25.7 days. In Schnitzler syndrome, the efficacy and safety of canakinumab were demonstrated in a randomized, placebo-controlled trial [38], and the 4-year extension study confirmed its sustained effects [39]. The interval between injections in the latter study was 62 days.

Canakimumab has the advantage of requiring fewer injections than anakinra. However, in the event of complications, the short half-life of anakinra is an advantage, as it is completely cleared within 48 h. In addition, canakinumab is an expensive treatment. In France, the cost of 150 mg of canakinumab is EUR 11,364, compared to EUR 32 for 100 mg of anakinra (i.e., EUR 1920 for 60 daily injections of anakinra).

#### 5.1.3. Rilonacept

Rilonacept is a chimeric recombinant fusion protein combining the extracellular ligand binding domain of IL-1R and the IL-1R accessory protein (IL-1RAcP). It acts as a soluble decoy receptor that binds to IL-1α and IL-1β. Rilonacept was approved by the FDA in 2008 for CAPS and in 2021 for recurrent pericarditis. The efficacy of rilonacept was reported in an open-label study that included eight patients treated with a loading dose of 320 mg followed by a weekly dose of 160 mg for 1 year [40].

### 5.2. Other Treatments

#### 5.2.1. Colchicine

Colchicine at approximately 1 mg per day may be effective in a subset of patients and may be considered as a first-line treatment in patients with mild disease and no persistent elevation of inflammatory markers, or as an adjunct in patients who do not fully respond to IL-1 inhibition [36].

#### 5.2.2. Tocilizumab

Tocilizumab is a humanized anti-IL-6R monoclonal antibody used primarily for the treatment of rheumatoid arthritis, giant cell arteritis, and juvenile idiopathic arthritis. The efficacy of tocilizumab was evaluated in an open-label study involving nine patients treated with weekly subcutaneous injections of 162 mg [41]. Tocilizumab was associated with a clinical and biological response in most patients, but a loss of efficacy was observed over time. It may be considered alone or in association in the rare cases of Schnitzler patients who do not respond to IL-1 inhibition [41].

#### 5.2.3. Ibrutinib

Ibrutinib is an irreversible inhibitor of Bruton’s tyrosine kinase (BTK) approved for the treatment of several lymphoproliferative disorders, including Waldenström’s macroglobulinemia. BTK has been shown to be a regulator of the NLRP3 inflammasome, and its blockade by ibrutinib has been shown in vitro to reduce IL-1β release by immune cells [42]. Partial or complete efficacy of ibrutinib has been reported in a few case reports [42,43,44].

## 6. Follow-Up

Schnitzler syndrome is a recurrent/chronic disease, and only one case of complete and prolonged spontaneous remission has been reported in the literature [45].

Serum AA protein (SAA) is a protein produced during inflammation that can form insoluble fibrils that accumulate in tissues. AA amyloidosis resulting from these pathogenic amyloid fibrils is a rare complication of chronic inflammation. It has been described in most subtypes of CAPS [46], and a few cases have been described in patients with Schnitzler syndrome after several years of untreated symptoms [2,47,48]. Interleukin-1 inhibitors are highly effective in preventing AA amyloidosis. Light-chain (AL) amyloidosis, which is secondary to the deposition of amyloid fibrils derived from the light chain of monoclonal immunoglobulin, has not been reported so far in patients with Schnitzler syndrome.

The major complication of Schnitzler syndrome is the development of lymphoproliferative disease, most commonly Waldenström’s macroglobulinemia. Due to the rarity of the disease and the lack of published follow-up data, the exact frequency of progression is unknown. The development of a hematologic malignancy has been reported in 35 of 281 patients (12%) with a median follow-up of 8 years after disease onset, but the true frequency is probably higher [2]. The risk is likely to be similar to the risk of progression in IgM monoclonal gammopathy of undetermined significance (MGUS). Most data suggest that suppression of inflammation with IL-1 or IL-6 inhibitors does not affect the monoclonal gammopathy and does not prevent the development of lymphoproliferative disease. Long-term follow-up of IgM gammopathy in patients with Schnitlzer syndrome is needed.

## 7. Conclusions

Schnitzler syndrome is an unusual late-onset acquired autoinflammatory syndrome. It must be considered in patients with refractory chronic urticarial rash associated with IgM monoclonal gammopathy. The rash in Schnitzler syndrome differs from classic urticaria by the absence of edema, the absence of pruritus, and its neutrophilic histology. Although suspensory, treatment with IL-1 inhibitors has dramatically improved the quality of life of patients but has no effect on the monoclonal component. Knowledge of the physiopathology of Schnitzler syndrome is expanding, but the main question regarding the relationship between the autoinflammatory features and the monoclonal gammopathy remains to be answered.

## Figures and Tables

**Table 1 biomolecules-14-00646-t001:** Lipsker diagnostic criteria of Schnitzler syndrome [33].

Urticarial rash and monoclonal IgM component and at least 2 of the following criteria ^1^:
Fever Arthralgia or arthritis Bone pain Palpable lymph nodes Liver or spleen enlargement Elevated ESR Leukocytosis Abnormal findings on bone morphologic investigations

^1^ In patients treated with IL-1 inhibitors, a rapid and immediate response is supportive of the diagnosis. In case of unresponsiveness to anakinra, the diagnosis should be reconsidered.

**Table 2 biomolecules-14-00646-t002:** Strasbourg diagnostic criteria of Schnitzler syndrome [34].

Obligate criteria: Chronic urticarial rash and Monoclonal IgM or IgG Minor criteria: Recurrent fever (>38 °C, otherwise unexplained) Objective findings of abnormal bone remodeling with or without bone pain ^1^ Neutrophilic urticarial dermatosis on skin biopsy Neutrophils > 10,000/mm^3^ and/or CRP > 30 mg/L
Definite diagnosis if the following are present: Two obligate criteria AND at least two minor criteria if IgM and three minor criteria if IgG Probable diagnosis if the following are present: Two obligate criteria AND at least one minor criterion if IgM and two minor criteria if IgG

^1^ Assessed by bone scintigraphy, MRI, or elevation of bone alkaline phosphatase.

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
