# Peer review of "Schnitzler Syndrome: Insights into Its Pathogenesis, Clinical Manifestations, and Current Management"

_biomolecules, 2024, doi:10.3390/biom14060646_

Round 1

Reviewer 1 Report

Comments and Suggestions for Authors

The review decribes what is known until now about a Schnitzler syndrome, which origin is unkonw. Schnitzler syndrome is characterized by a chronic urticarial rash and  monoclonal IgM (or IgG) gammopathy with other  secondary  sindromes . It seems that this phatology is promoted  by a defective inflammasome. The authors report the major informations about the disease present in litterature. References are adeguate.  The review is well organized. English is good.  As there are few studies about this rare desease I sugest to the authors:

A) line 51: introduce a figure, if it is  possible

B) line 123:  A recent hypothesis on the involvement of B clonality in abnormal inflammasome activation is present in literature:  autoantigens which may recognize by IgM (dipeptidyl peptidase 10 and efflux potassium mediated NLRP3 activation.) It may be mentioned in the review.

C) line 123: May be overproduction of IL-18 responsible of B cell expansion and/or growth? Introduce possible explanation/ hypothesis

D) line 194: Contextualize bone abnormality, extensive described in your article, respect to some cytokines overproduction in Schnitzler syndrome and their possible involvement [CCL2, IL-1, IL-6 and IL-18]

E) line 179: Neutrophils and NETosis are involved in many autoimmune. Also, in skin of Schnitzler syndrome patients an infiltration of neutrophiles is described. What is known   about involvement of both neutrophils and NETosis in Schnitzler syndrome?

Author Response

Thank you for taking the time to review this manuscript. Please find below the detailed responses. The changes are highlighted in the manuscript with track changes in the resubmitted files. All line refer to the revised manuscript file with tracked changes, showing simple markup.

Comment A) line 51: introduce a figure, if it is possible

Response: Thank you for this suggestion. However, numerous figures illustrating the NLRP3 inflammasome have already been published and this is slightly outside the scope of this review. Therefore, we have not included such a figure, but we have included references to key articles dealing with the NLRP3 inflammasome.

Comment B) line 123:  A recent hypothesis on the involvement of B clonality in abnormal inflammasome activation is present in literature:  autoantigens which may recognize by IgM (dipeptidyl peptidase 10 and efflux potassium mediated NLRP3 activation.) It may be mentioned in the review.

Response: Thanks for pointing this out. DPP10 was indeed mentioned by Pathak et al. (PMID: 33424831) when they investigated the role of the IgM paraprotein in the pathogenesis of Schnitzler syndrome. DPP10 was identified using a protein microarray performed on total IgM from three Schnitzler patients and two Waldenström patients (as a negative control for autoinflammatory features). However, while it might be physiologically hypothesized that the IgM-DPP10 interaction could alter NLRP3 function, DPP-10 was identified as a putative interactor in all 5 samples (i.e. patients and controls) using the authors' threshold (z-score > 2.5). We agree with the authors' conclusion that their results from both high-throughput sequencing of the IgH region and the IgM protein microarray do not support the notion of an antigen-driven pathogenesis for Schnitlzer syndrome. Therefore, although this study is referenced in our manuscript, we have decided not to elaborate on this DPP10 hypothesis.

Comment C) line 123: May be overproduction of IL-18 responsible of B cell expansion and/or growth? Introduce possible explanation/ hypothesis

Response: We mentioned IL-6 in our manuscript because IL-6 is recognized as a proliferative factor in plasma cell dyscrasias and in particular in Waldenström's disease. Indeed, IL-6 has been shown to promote cell growth and IgM secretion by interacting with tumor cells. IL-18 is indeed activated during NLRP3 inflammasome activation after being cleaved by caspase-1. The role of IL-18 in cancers appears to be context-dependent, as both pro- and anti-tumor effects have been reported. However, we are not aware of any strong evidence regarding the role of IL-18 in the natural history of Waldenström's disease and therefore did not include this hypothesis in our manuscript.

Comment D) line 194: Contextualize bone abnormality, extensive described in your article, respect to some cytokines overproduction in Schnitzler syndrome and their possible involvement [CCL2, IL-1, IL-6 and IL-18]

Response: We have accordingly added more information on the hypothesized pathogenesis of bone lesions (line 239 -243)

Comment E) line 179: Neutrophils and NETosis are involved in many autoimmune. Also, in skin of Schnitzler syndrome patients an infiltration of neutrophiles is described. What is known   about involvement of both neutrophils and NETosis in Schnitzler syndrome?

Response: Thank you for pointing this out. We agree that some information on neutrophils in Schnitzler syndrome was missing. Accordingly, we have added a section on neutrophil infiltration and NETosis in Schnitzler syndrome (lines 105-123).

Reviewer 2 Report

Comments and Suggestions for Authors

Schnitzler Syndrome: insights into its Pathogenesis, Clinical Manifestations and Current Management by Braud et Lipsker, is an interesting and pertinent compilation of information about this disease. This syndrome could be considered a rare disease and is associated with gammopathy. As indicated, this disease presents characteristics similar to other autoimmune diseases. The article presents a very summarized introduction, referring in different chapters to the pathogenesis, epidemiology (which due to the information it presents should be considered part of the introductory chapter 1), clinical diagnosis, therapeutic management and some perspectives suggested in chapter 7. which is unclear, the information indicates in relation to amyloidosis it would be interesting if handled as a section on differential diagnosis and the comment in this section that they should be better focused on trying to propose options for diagnosis, prognosis. The work is interesting and would be of potential interest to readers of “Biomolecules”, but it is necessary for the authors to reorganize the way of presenting the manuscript. The organization of the manuscript into chapters and subchapters should be reconsidered. For example, Chapter 2 on pathogenesis does not identify the need to prepare sub-chapters, especially that subchapters 2.1.3 and 2.1.3 (2.1.3 No evidence for NLRP3 mutation in Schnitzler syndrome is it an error that the numbering is repeated?) deal with to be related, although as mentioned in No evidence for NLRP3 mutation in Schnitzler syndrome (this section should be part of the previous paragraph). Similarly subchapters 2.1.4 and 2.1.5 should be a single section.

Author Response

Thank you for taking the time to review this manuscript. Please find below the detailed responses. The changes are highlighted in the manuscript with track changes in the resubmitted files. All line refer to the revised manuscript file with tracked changes, showing simple markup.

Schnitzler Syndrome: insights into its Pathogenesis, Clinical Manifestations and Current Management by Braud et Lipsker, is an interesting and pertinent compilation of information about this disease. This syndrome could be considered a rare disease and is associated with gammopathy. As indicated, this disease presents characteristics similar to other autoimmune diseases.

Comment: The article presents a very summarized introduction, referring in different chapters to the pathogenesis, epidemiology (which due to the information it presents should be considered part of the introductory chapter 1)

Response: We completely agree, the epidemiology has been moved to the introductory chapter.

Comment: , which is unclear, the information indicates in relation to amyloidosis it would be interesting if handled as a section on differential diagnosis and the comment in this section that they should be better focused on trying to propose options for diagnosis, prognosis.

Response: The relationship between Schnitzler syndrome and both AA and AL amyloidosis has been clarified in the prognosis section (line 391-399). Following your suggestion, the differential diagnoses have been individualized in a subchapter with more detail regarding the key differences with Schnitzler syndrome (line 292-308)

Comment: The work is interesting and would be of potential interest to readers of “Biomolecules”, but it is necessary for the authors to reorganize the way of presenting the manuscript. The organization of the manuscript into chapters and subchapters should be reconsidered. For example, Chapter 2 on pathogenesis does not identify the need to prepare sub-chapters, especially that subchapters 2.1.3 and 2.1.3 (2.1.3 No evidence for NLRP3 mutation in Schnitzler syndrome is it an error that the numbering is repeated?) deal with to be related, although as mentioned in No evidence for NLRP3 mutation in Schnitzler syndrome (this section should be part of the previous paragraph). Similarly subchapters 2.1.4 and 2.1.5 should be a single section.

Response: Thanks for pointing this out. We have indeed made some mistakes with the numbering of chapters and subchapters. The chapters have been renumbered and reorganized according to your suggestions.